# Managing Global Smart Cities in an Era of 21st Century Challenges

**Milan Kubina** **, Dominika Šulyová * and Josef Vodák**

Faculty of Management Science and Informatics, University of Zilina, Univerzitna 8215/1, 010 26 Zilina, Slovakia; milan.kubina@fri.uniza.sk (M.K.); josef.vodak@fri.uniza.sk (J.V.)
* Correspondence: dominika.sulyova@fri.uniza.sk; Tel.: +421-41-513-4022

**Abstract:** Globalization, integration and liberalism are concepts that have been used since ancient history and have influenced urban governance to this day. The aim of the article is to find out, based on the historical development of globalization, Friedmann's urban concept and Sassen's global theory of cities—how world cities reflect the new challenges of 21st century globalization. In the recent past, building of the global urban network has been influenced by factors such as the growth of populism, neoliberalism, migration, the existence of exploitative centers, urbanization and changes in the demographic curve. Similar to the year 2020, also in 2021 cities must face a single global challenge posed by the Covid-19 pandemic. In this article the authors used methods of comparative analysis of global Smart Cities such as New York, London and Tokyo. The discussion section includes a summary of results of the analysis, and a design of a new general model for managing global challenges in cities is introduced. The results of the article point towards the role and influence of cultural differences of global cities and this also relates to the approach to managing the new challenges of current times. New York and London are culturally closer and also showed similar results, whereas Tokyo differs across all analyzed elements. The main result of the article are the answers to the research questions and the design of a new general model which involves various elements of globalization management and which is based on the world best practices.

**Keywords:** globalization; Smart City; global city; world city network; management

## 1. Introduction

Globalization, the unifying process connecting activities, countries and resources into a complex whole, began to develop at the time of the first settlement in the world and continues to the present day of the 21st century. Society has in the meantime evolved from the days of food migration, the agricultural revolution, the Silk Road trade, and the industrial revolution to the scientific revolution [1].

The end of the 18th and the beginning of the 19th century represent the first wave of globalization, which is characterized by rapid development of Smart Cities. The globalization of the 20th century is also known as the "age of responsibility", the main aim of which was to maintain peace, stability and democracy after the Second World War [2]. The third wave of globalization lasted until 2008 on the basis of the global supply chain using internet services.

The most recent wave of globalization represents the current state of the 21st century, known as "Globalization 4.0" or the "Age of Revolt" [1,2].

The purpose of the article is to propose a general model for managing global challenges faced by the Smart Cities, based on a comparison of the best practices of global cities such as New York, London and Tokyo. The aim is to identify, across the globe, common as well as different elements of globalization management used in practice worldwide, using methods of secondary analysis, comparison and summarization. Three research questions were formulated for the purposes of the article:

- What are the common and different elements of globalization management in New York, London and Tokyo?
- What elements influence the management of global cities today the most?
- What is the most appropriate way to manage global challenges within the concept of global Smart Cities in general?

In order to understand the issue studied in the article, it is important to create an operational definition of the term globalization through existing selected definitions, a summary of which can be found in Table 1.

**Table 1.** Defining the term globalization.

| Author | Globalization |
| --- | --- |
| Aron, 1968 [3] | The current process influenced by technological and economic factors, which is to lead to the unification of humanity. |
| Modelski, 1972 [4] | A historical process that is not possible to control or manage. |
| Lewit, 1983 [5] | Current, irreversible and standardized process of unification of products and countries. |
| Beck, 2004 [6] | Globalization is a model of risk management based on the principle of universalism. |
| Walterstein, 2005 [7] | The concept of globalization is just an invention of power elites who abuse it as an ideological weapon. |
| Naím, 2008 [8] | Rapidly advancing integration of economies, cultures and nations. |
| Kozárová, 2013 [9] | A social phenomenon that builds collective consciousness, a historical, manageable process of an iterative and contradictory nature, influenced by the driving forces of development. |
| Ejal, 2020 [2] | Globalization is not a natural phenomenon, but a consequence of the economic and political situation, which has increased the quality of life but disrupted communities and the ecosystem. |

Clearly, there are different ways of thinking about and understanding globalization, for example, seeing it as a historical process as proposed by Modelski and Kozárová, or seeing it more negatively and interpret integration as a power struggle as proposed by Walterstein. A common aspect is unification and integration. In general, one may argue that the negative aspects of globalization are beginning to have a greater impact than the benefits that the process can generate.

Within the operational definition, globalization will be perceived as an integrated consequence of the current situation of the 21st century, with the benefit of increasing the income of higher social classes, characterized primarily by negative effects on the global ecosystem and the society in the form of populism, social inequality and the existence of exploitative centers.

### 1.1. Recent Past and Present of Globalization

The events of the recent past, from 2016 to 2020, introduced fundamental challenges facing all countries of the world. In addition to the Covid-19 pandemic, the growing popularity of populism, illiberalism, anti-globalization, migration, the existence of exploitative centers and a negative perception of globalization, all pose global problems. Representatives of the strategic management of cities, countries and nations in 2021 [2] must face all of these challenges.

1.1.1. Popularity of Populism, Neoliberalism and Anti-Globalization

Middle working class, so-called blue collars, has low confidence in the state and a negative perception of globalization based on the loss of national identity, values and communities [2]. The age of responsibility was replaced by a revolt, which began with a terrorist attack on World Trade Center on 11 September 2001 [2]. Since 2016, when Donald Trump became President of the United States, a dichotomy has emerged that argues that the certainties and ideas of the 20th century are no longer popular. According to Trump, globalization of the 21st century is a term respected by people who "care about the well-being of the world, but are not interested in their own country" [2].

Research from 2019 confirmed that the correlation between trust and innovation prevails. If citizens trust the state, innovations are encouraged, which causally results in a

higher quality of life [2,10,11]. Americans' distrust deepened, especially between 1997 and 2007 [12]. The problem of pollution of the Flint River by General Motors, which top management has not been able to tackle effectively, has also led to a rapid decline in confidence [2]. The causal consequence of this was the decline of trust among Americans. On a scale of 1 to 10, citizens trust in other people and institutions reached only 5.8 points [13].

Progress and sustainable development can only work on the basis of rational policy, protection of limited resources, transparency of information and trust [2].

### 1.1.2. Migration

In 2019, London ranked in several world rankings as the leading Smart City, which is the best practice for diversity management and multiculturalism [14–18]. According to Griffin, diversity is not governed by open access, as Londoners did not receive transparent information and separation of the upper classes was observed. The dissatisfaction resulted in the UK's Brexit from the European Union in 2020 [2].

Historically, survival of a population depends on its ability and right to migrate. Migration dynamically contributes to the change and development of a civilization. From an economic point of view, it brings innovation, employment and productivity growth [2]. According to a research by the International Monetary Fund, there is a correlation between the number of migrants and the improvement in the quality of life [19]. In the recent time period, migrants have been leaving their homes mainly due to the conflicts in the Middle East region [2].

Movements of the migrants also represent certain challenges, for example, loss of national identity, the basic iterative problem of globalization, the contradiction between the universal and the specific, the global and the local [2].

### 1.1.3. Exploitation Centers

The existence of exploitative cities has a negative impact on the living conditions of the population and the state of the environment. Beijing is a city known for its power plants and quality industrial products. However, business profit creates an externality associated with a high degree of air pollution, which causes death of approximately 4.2 million people, mainly in China. The indicator of wealth in Beijing in 2017 was not the number of cars or the amount of money in one's bank account, but the ownership of the air purifier. Based on the complaints from the city residents, power plants and factories were moved to rural areas, that is, the poorest regions of China [2]. Citizens of Beijing, according to the so-called "Air Quality Index" created by the University of Chicago, have shorter life span by on average 2.3 years due to poor air quality. On the one hand, Beijing is a technological and industrial power, on the other hand it is also known as a center of exploitation [2]. In 2018 China was one of the most efficient waste processors in the world. Western countries moved unwanted material to the eastern region, creating a mediated negative externality [2].

The main consequences are environmental pollution, lack of limited resources and unsustainability of current ecosystems for future generations. The governmental solutions are insufficient, short-term oriented and transfer shortcomings to other areas, rapidly reducing confidence in state institutions [2].

### 1.1.4. Current Situation in 2021

The Covid-19 pandemic highlighted the positives and negatives of globalization. Transmitting problems, in this case a contagious virus, across borders in conjunction with finding common solutions, sharing knowledge and material is a paradigm of today. The initially regional impact of the virus in China spread to become a global problem, mainly as the Chinese government sought to address it in isolation [2]. Nature can no longer recover from the damage caused by the era of industrialization and technological development. The negative externality has thus created the exponential problem of Covid-19, which is one of the global challenges that require global solutions [2].

People are increasingly rebelling against globalization as it benefits only some members of the upper social class [2]. According to a 2017 research report from Oxam, the following facts emerged [20]:

- The eight richest people in the world generate more income than half of the globe.
- In the United States, the income of the poorest strata has not risen at all; the income of the social elite has risen by up to 300%.
- The richest entrepreneur in Vietnam earns per day as much as one poor person does in 10 years.

World Social Report statistics from 2020 confirm the unequal distribution of income. It will take 40 years of effort to harmonize wealth between social classes. Additionally, as a result of climate change, social inequalities are deepening by up to 25% [21].

### 1.2. Global Theory of Smart Cities

The connection between globalization and building of Smart Cities was realized by creating a paradigm in the 1980s through Friedmann's world urban model and Sassen's global theory of cities [22].

Friedmann and Wolff argued that the global aspect of cities manifests itself in the form of power and centralized management of businesses, financial institutions and infrastructure. Smart Cities are connected globally through the managerial function of decision-making, control and funding. This approach extends Taylor's theory to "A network of collaborating cities around the world" [23–27].

Friedmann's model of sustainable cities with peripheral cores comprises four phases [28]:

- Agricultural society—is characterized by low mobility and regional differences based on the natural advantage of a particular location.
- Transitional—the economy and innovation are concentrated in the city center, which is the focal point of commercial and industrial activities typical of the first and second industrial revolution.
- Industrial phase—based on the growth of costs and integration processes of the infrastructure in the city center, new areas of growth are emerging on the principle of deconcentration.
- Post-industrial phase—a fully integrated system supporting globalization.

A comparison of urban development models by Friedmann, Gibbs and Hautamäki (according to Raagmaa) created a new peripheral model based on the principle of synthesis. It consists of five phases [29]:

- Initial phase—agrarian society with small towns.
- Local urbanization—support for industrial production, technology in the center of the capital. People's mobility is low.
- Urbanization in the center—people are moving to the capital for work and better living conditions. However, a critical factor in success is the stabilization of agriculture on the outskirts of the city.
- Sub-urbanization and creation of new cities—with ever-increasing mobility, industry, innovation, growth and development of cities and regions.
- Urban sprawl—cities are becoming global agglomerations, urban centers are stagnating as problems are generated with transport, crime, environmental pollution and high housing costs.
- Counter-urbanization—the mobility of inhabitants to large cities is rapidly declining due to the application of trends such as changes in the demographic curve, unemployment due to migration, lack of limited resources, etc., which increases the importance of regional and rural areas.

By synthesizing the peripheral model, new elements of urban development have been added to economic, political factors, technologies and cultural aspects, such as [29]:

- globalization,
- services,

- tourism,
- smart products,
- renovation of old city centers,
- digitization, robotics.

However, continuously generated agglomeration problems and trends are a persistent factor [29]. An associated model of systems development is the iterative model from Chase-Dunn. Population growth affects the so-called "Intensification", that is, higher consumption of limited resources such as water, soil, energy and air. Their lack leads to environmental degradation and lower food production, which negatively affects pressure to the population, which results in migration activities. This naturally creates a circle or a limit at which people want to live as before, but the impact of population pressure will result in the acceptance of change, innovation and technology. If this impact is long-term, it can cause conflicts [30].

When wars and riots influence this pressure, the system enters a "vicious circle." Under positive contextual conditions, conflicts can generate hierarchical structures. The adoption of centralized power will be reflected in new technologies in the management of semi-peripheral countries [30].

Sassen's global theory of cities reflects Friedmann's model. Sassen argues that global Smart Cities manage the world economy. It is important to focus on practical control, synergy and the cluster, and not the formal strength represented by the number of municipal enterprises [31–34]. According to Sassen, the global cities include three world agglomerations, New York for North America, London for Europe and Tokyo for Asia, the so-called triumvirate. These cities thrive because they form financial and management centers based on the managerial function of control and cooperation of companies [31–34]. According to Sassen, central urban management is a critical success factor. The difference between Sassen's theory of cities and Friedmann's model is an argument that Friedmann did not assume. If companies at the global city level only sell products to each other, can the situation be considered a measure of global control or even cooperation? In her publications, Sassen argues that the key to urban success is not to describe cooperative contexts, but to focus primarily on global challenges and external relations between cooperating cities, which is the main difference between Sassen's approach from Friedmann. Global cities thus form the primary development point of the economy as an urban network based on cooperation [31–34].

### 1.3. The Impact of Globalization on Management

Development trends and technologies affect managerial functions and organizational management processes. Global challenges can only be met through education and knowledge management.

Strategic management decisions should favor diversity, multiculturalism of lifestyle, that is, stabilizing the differences between local and global. The macro environment influences management mainly in the form of trends (urbanization, technologies, changes in the demographic curve, etc.), which have a significant impact on structures, strategies and development plans. Managers play a key role in the age of globalization. Their decisions, communication approaches and strategies generate a global image (reputation) and competitiveness of a country [35].

In the context of effective globalization management, strategic management should focus on six key areas by Harvey et al. [36]:

- Decision-making through global thinking—in addition to rationalization, it is necessary to take into account local conditions, strategy, power, traditions, but especially intuition, emotional intelligence, openness, socio-psychological aspects, and the ability to risk or reach consensus.
- Knowledge management—the collected data need to be transformed into information to support management and decision-making, and then into knowledge as a competitive advantage of management.

- Technologies—analysis of the current and future state of innovation in several areas.
- Dynamic adjustment of strategy, time and capacity based on changing market conditions and trends.
- Cooperation—creation, stabilization and development of international cooperation.
- Organizational structure—prefer flexible forms of hierarchy.

The fundamental process of globalization between the individual and the collective, universal and specific, intervenes in the governance process [36].

## 2. Materials and Methods

The selection of Smart Global Cities to perform a comparative analysis was performed through a secondary analysis of the literature from various experts and their views on the best global practice of cities between 1991 and 2020, which can be found in Table 2. Only significant milestones from 1991, 2002, 2010 and 2020 were selected.

**Table 2.** Global cities according to experts.

| Authors | Area/City |
|---|---|
| Sassen, 1991 [32] | Asia—Tokyo |
| Hoyler, Pain, 2002 [37] | Europe—London |
| Brown et al., 2002 [38] | North America—Miami |
| Bassens et al., 2010 [39] | Middle East—Tehran, Manama, Dubai |
| Global Cities Index 2020 | |
| Kearney, 2020 [40] | 1st place—New York<br>2nd place—London<br>3rd place—Paris<br>4th place—Tokyo |

Source: own processing by the authors, according to professional literature [41].

Table 2 and Kearney's 2020 surveys show that the best practice for building global cities is for America's Smart City New York, for Europe's London and Paris (the article will focus on higher-ranking London), and for Asia it is Tokyo; that confirms Sassen's 1991 argument that the best example of a globalized city is Tokyo [42].

In addition to the secondary analysis of the literature and case studies, the article also used methods of comparing the results of three selected sites. A summary of the results of the comparison is given in the discussion section.

Tables 3 and 4 summarize, based on the results of a comparative analysis of the selected three cities (New York, London and Tokyo according to Table 2). The common and different management elements of these global cities has been selected on the basis of analysis and comparison of practical studies by Sassen, Kantor. et al., and Hill and Kim, who described a combination of global Smart Cities management (New York, London and Tokyo) and specific studies from Saito (for New York and Tokyo) and Bacon (for London) [32,43–46].

**Table 3.** Common elements of globalization management.

| Common Elements | New York | London | Tokyo |
|---|---|---|---|
| Center for growth and social diversity. | Yes | Yes | Yes |
| Existence of separation of society classes and social inequalities. | Yes | Yes | Yes |
| The impact of global trends and problems of globalization on city management. | Yes | Yes | Yes |

Source: own processing by the authors, according to Section 3 [43–46].

The results of the theoretical part of the article (Section 1) and the comparative analysis (Section 3) served as a basis for answers to three research questions:

- What are the common and different elements of globalization management in New York, London and Tokyo?
- What elements influence the management of global cities today the most?
- What is the most appropriate way to manage global challenges within the concept of global Smart Cities in general?

The main output of the article are answers to three research questions and a general model for managing the global challenges of Smart Cities, which is part of Section 4.

**Table 4.** Different elements of globalization management.

| Different Elements | New York | London | Tokyo |
|---|---|---|---|
| Global Financial Center. | Yes | Yes | No |
| Center of economics, culture, and tourism on a global level. | Yes | Yes | No |
| Globalization is driven by capitalists, the private sector and the market. | Yes | Yes | No |
| Focus on the tertiary services sector. | Yes | Yes | No |
| The employment structure is polarized, with a disappearing middle class, a high level of separation of social classes and social inequalities. | Yes | Yes | No |
| Migration is high and its control is insufficient. | Yes | Yes | No |
| City management is focused on integration. | No | No | No |
| Centralist management prevails. | No | No | No |
| Mobility and flexibility are the main competitive advantage of the city. | Yes | Yes | No |

Source: own processing by the authors, according to Section 3 [43–46].

## 3. Results

*Comparative Analysis of the Global Cities of New York, London and Tokyo*

New York is one of the centers of growth, finance and social diversity. Global challenges are managed through a capitalist approach, that is, market-oriented management and the tertiary services business sector. Migration is on the rise, but from a managerial point of view it is not subject to strict control. Smart City New York is influenced by separation of society classes, the middle class revolt (Section 1.1.1) and decentralization.

In the period from 2016 to 2020, politics were dominated by the Donald Trump's anti-global approach. Currently in 2021, John Biden was elected President; he promotes democracy and thus the age of responsibility and globalization. The solution to the challenges in America is ambiguous, as it is characterized by a fragmented society at the interface between democracy and populism [43–45].

London manages the aspects, impacts and challenges of the globalization process using three levels [46]:

- Strategic level—the Greater London Authority's projects create environmental projects.
- Communities—government authorities support education, change of mindset to positively embrace technology, or strategies and projects to protect limited resources.
- Local level—diversity management is implemented through projects for the creation of cultural centers, for example, "Rich Mix", support for the Chinese business "Emerald Center", and the creation of a Muslim center or the involvement of Asian children in sports activities.

The capital of the United Kingdom has a similar focus on global management as New York (Tables 3 and 4). The essential difference is the response to migration. The Americans tried to solve this transnational challenge with new management that would reflect their views (Trump's protectionism, trade war with China, the wall near Mexico, etc.). The British took a more radical step, i.e., Brexit and protectionism, which Japan has preferred since the 20th century [45].

One of Tokyo's unique elements is its focus on industrial production. On the other hand, the inflow of foreign direct investment is low compared to other global cities such as New York and London. In Japanese cities, the trend of higher emigration than immigration has prevailed since 1980. The globalization aspects of Eastern countries are thus reflected

in the export of people, products and investment, including restrictions on imports. Social and income inequality was also reflected in Smart City Tokyo, but was lower than global standards [45].

In New York and London, positions in the tertiary services sector were better paid. Migrants who moved to Tokyo held positions in lower-paid jobs in services, small manufacturing companies and construction. Their placement in these jobs, the so-called fragmentation did not have a negative impact on the labor market. The lower level of social polarization created a stable and safe space for life [45].

Globalization in Tokyo is managed mainly through the state and ministries, while in London and New York it is the private sector and the market [45]. A summary of the common and different elements of globalization management of selected Smart Cities can be found in Tables 3 and 4.

## 4. Discussion

The results of secondary literature analysis and case studies provided data for answers to the three research questions.

### 4.1. What Are the Common and Different Elements of Managing the Globalization Aspects in New York, London and Tokyo?

Summaries of common and different elements of globalization management in the analyzed cities can be found in Tables 3 and 4.

The biggest differences in Tables 3 and 4 are between Western cities and Tokyo. Unlike New York or London, Tokyo is not a global financial and cultural center, and globalization is managed centrally through the state [43–45].

The employment structure is absent from extremes, with lower levels of separation of social classes and social inequalities. The number of migrants is lower as migration is strictly controlled. The stability and planning of Smart City Tokyo is a competitive advantage [43–45].

### 4.2. What Elements Influence the Management of Global Cities Today the Most?

According to Harvey (Section 1.3), the global management of Smart Cities is influenced by the managerial functions of planning, control and management, the level of which depends on the level of intelligence quotient (IQ), and the management of tasks at the set time. These aspects are reflected in the hierarchy of the system, that is, governance, trends in new technologies and migration, population and environmental issues (Section 3 and Tables 3 and 4).

### 4.3. What Is the Most Appropriate Way to Manage Global Challenges within the Concept of Global Smart Cities in General?

Based on the findings of Friedmann's and Sassen's theory of cities and models of Gibbs, Hautamäki (according to Raagmaa) and Chase-Dunn and the results of a comparative analysis in Section 3, it was possible to construct a graphical representation of effective management of global challenges according to the world's best practice Smart Cities.

The central element of the model according to the results in Section 3 and by Saito is the core of Smart City, which consists of financial, commercial and production centers of a specific city. Saito claims that New York and London have financial and business centers at their core, and Tokyo prefers a production focus (Table 4).

According to Sassen and Ejal (Sections 1.1 and 1.2) trends in urbanization, mobility and innovative development are at the core. The center maintains mutual relations with adjacent city districts (opinion by Sassen's theory in Section 1.2), thus forming the so-called urban network based on cooperation (Figure 1).

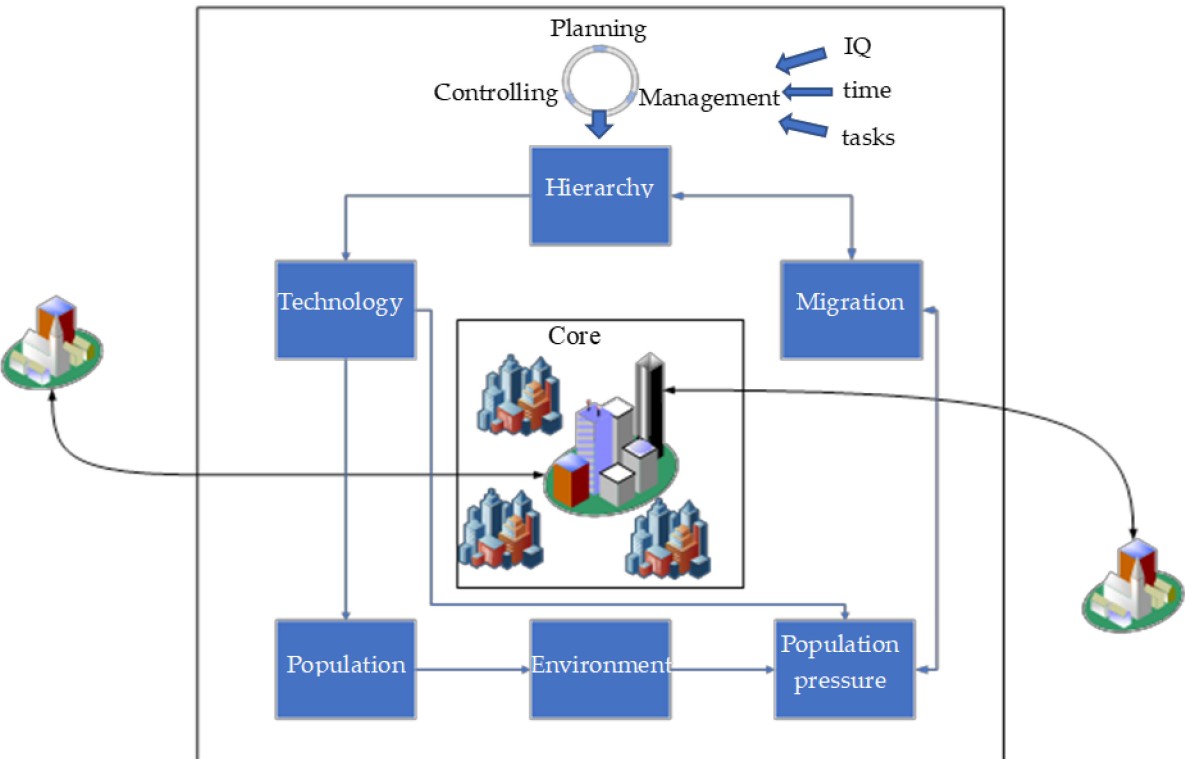

**Figure 1.** General globalization/global challenges management model for the Smart City (own processing by the authors according to the results of literature review and case studies).

Elements of globalization are concentrated around the core. For effective management of global challenges the model contains, as described by Friedmann´s model (Section 1.2), managerial functions of planning and management. By Harvey et al. management depending on intelligence quotient (IQ) of managers (analytical, creative and practical level of intelligence), fulfillment of tasks in a set time and continuous control to modify or improve the current Smart City management processes.

As described by Chase-Dunn and Harvey et al. (Sections 1.2 and 1.3) the cycle (Figure 1) affects the formal hierarchy of the city, which has a positive impact on the adoption and implementation of new technologies. Innovation, applications and technologies increase mobility to cities, generating more population. The results of comparative analysis (Section 3) indicates that according to Tokyo it is appropriate to manage the city through a managerial control function.

Rapid population (by Chase-Dunn in Section 1.2) growth has a negative impact on the environment (according to results of London in Section 3) and causes a loss of limited resources. Adaptation to change, new technologies and scarcity of resources, including pollution, increase the level of population pressure. The consequence is the interrelationship between the pressure and the migration aspect. The lower the management pressure on the population, the greater the migration to the site. If the pressures are too high, the population will emigrate to other places with lower population pressure.

The relationship in Figure 1 represents the feedback between migration that affects hierarchy (management) and vice versa. Prediction-based model integration is required.

From the comparative analysis, critical factors of success were included in the model, which in the opinion of the authors should be implemented in the general model of globalization management/global challenges for the Smart City area (Figure 1). These elements include (Tables 3 and 4):

- management of global challenges, including technological development, changes in the demographic curve, migration (Section 1.1.2), environmental pollution (Section 1.1.3) and the style of the management hierarchy (1.3.),

- the financial, commercial or production core of the city,
- interconnection between the internal core and the external urban areas,
- flexible adaptation to change (New York and London in Table 4), which, however, must be based on a stable city center on a planning and predicate basis (Tokyo in Table 4),
- centralist governance of the city core (a common element of Friedmann's and Sassen's theory, including Tokyo in Section 3) in collaboration with business and the market (New York and London element in Table 4).

According to Portes' study from the year 2020 global cities should be governed by a predictable democratic political environment. The critical success factors are the creation of a reliable legal regime, effective management, business and financial transactions or the strategic distribution of the population according to local conditions. These elements are difficult to integrate into a single regional entity, which is a major challenge for future global Smart Cities [47].

## 5. Conclusions

The process of globalization has contributed to the dynamic development of the world economy, multiculturalism, technology and living conditions. In the conditions of the 21st century however, the population feels dissatisfied with the current economic system based on the principle of globalization, which they compare to the exploitation of the lower and middle social strata. The primary benefits of integration in the form of free trade, freedom of movement, the sharing of human capital and the integration of foreigners into local culture are now limited. Current theories are based on Friedmann's model and Sassen's theory of global cities, which are also important for the current responses of cities to the global challenges of the 21st century. Globalization has a major impact on governance and management. Effective governance should be based on the principles of rational decision-making, which takes into account and seeks to harmonize the contradictions between collective, universal, individual and local. Data collected by smart sensors and products should be processed into information within knowledge management. Dynamic adaptation to technological change, innovative forms of organization, strategies and time based on cooperation are critical factors in the success of global city management.

The results of a comparative analysis of the world's best practices of global Smart Cities, including New York, London and Tokyo, have highlighted the significant impact of culture on meeting global challenges by managing selected Smart Cities. New York and London had similar results, but comparative differences were seen in Tokyo management. Through a high degree of migration control, it achieves a lower level of separation of social classes and social inequalities. Management is centralized, industry is focused on production and not services, which generates a unique view of the city. Competitive advantage is preparedness, plans and integration. Based on the findings and results of the article, the key factors in the management of 21st century global cities are elements such as:

- flexible response to global challenges and trends,
- stabilization of the city center through the managerial planning function,
- prediction of the future state,
- managing a population explosion due to migration through its control,
- perception of cultural differences and diversity as benefits,
- seek to protect the environment and eliminate exploitative centers,
- prefer centrally oriented management in cooperation with the city's stakeholders.

The main goal of the article was achieved through the creation of a general model of globalization management and its challenges for the Smart City area, which consists of elements of managerial functions, hierarchy and global trends (migration, population growth, technology development or environmental pollution). Within the systemic interconnection of elements and interrelationships, it is important to highlight the factor of control and prediction of the future. Integration is the key word with respect to globalization. However, it is very difficult to connect all local elements into a single universal complex system. If

the city's management nevertheless succeeds, it will gain a competitive advantage and a developed Smart City space for future progress.

**Author Contributions:** Conceptualization, M.K., D.Š. and J.V.; supervision, M.K. and J.V.; formal analysis, D.Š.; methodology, D.Š.; writing—original draft, D.Š. All authors contributed to the manuscript preparation. All authors have read and agreed to the published version of the manuscript.

**Funding:** This research received no external funding.

**Institutional Review Board Statement:** Not applicable.

**Informed Consent Statement:** Not applicable.

**Data Availability Statement:** No new data were created or analyzed in this study. Data sharing is not applicable to this article.

**Conflicts of Interest:** The authors declare no conflict of interest.

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
