# Peer review of "Managing Global Smart Cities in an Era of 21st Century Challenges"

_sustainability, doi:10.3390/su13052610_

Round 1

Reviewer 1 Report

Dear authors, your paper is newsworthy. It sounds clear and effective. Very interesting the literature review, clearly linked with a strong theroetical background. However, materials and methods description appear to be not so detailed, considering that the proposed element of globalization in table 3 and 4 would have benefit of an appropriate and exaustive presentation about how have been selected, as it is thanks to them that it is possibile to give and answer to the three proposed questions.

Moreover the general globalization / global challenges management model for Smart City in figure 1 appears to be vague in term s of interlinkages if compared with the text. Sounds unclear the role of cited index in the preparation of the model.

Conclusion could benefit of broader and more exhaustive remarks.

Reviewer 2 Report

The article is interesting and very well written. I enjoyed reading it.

However, here are some observations that may help the authors improve their manuscript:

  • The authors should better explain in the “Discussion” the relationship between the elements of the cited theories/models, the results of the comparative analysis developed, and the model proposed; in other words, the sections considered in isolation have their own coherence, but the relationships between them do not appear evident and should be better highlighted.
  • In line 206, the Wolf model is mentioned, but the authors do not explain what it consists of and how it differs from the others.
  • Sassen's theory should be explained in more detail, as it is also cited in the abstract as a basis for the article, but the authors devoted to it much less space (about 5 lines) than Friedmann's model.
  • In line 317, the acronym IQ is used, but its meaning has never been explained
  • In line 232, the model of Kultalahti is cited as a basis for the findings, but in the literature review, it was neither explained nor cited.
  • In section 5.2, it is necessary to refer to the comparison between the three cities.

Finally, here are some formal notes:

  • The numbering of the sub-paragraphs must be reviewed (e.g. line 75, line 94, etc.).
  • It would be necessary to consider whether to replace the term "chapter" with "section" (e.g. 252).
  • In figure 1, the typo of the word environment must be corrected.
